# Pharmacological Effects of Grape Leaf Extract Reduce Eimeriosis-Induced Inflammation, Oxidative Status Change, and Goblet Cell Response in the Jejunum of Mice

**DOI:** 10.3390/ph16070928

**Published:** 2023-06-26

**Authors:** Mutee Murshed, Jameel Al-Tamimi, Hossam M. A. Aljawdah, Saleh Al-Quraishy

**Affiliations:** Department of Zoology, College of Science, King Saud University, P. O. Box 2455, Riyadh 11451, Saudi Arabia; jtamimi@ksu.edu.sa (J.A.-T.); haljawdah@ksu.edu.sa (H.M.A.A.); squraishy@ksu.edu.sa (S.A.-Q.)

**Keywords:** *Vitis vinifera*, eimeriosis, parasitic stages, jejunum, histology, goblet cells

## Abstract

Coccidiosis is a parasitic infection threatening poultry products globally. Parasite resistance to drugs is one of the barriers to *Eimeria* control. Natural products are one of the sources of compounds that prevent parasite infections. The current study was, therefore, conducted to evaluate the effect of *Vitis vinifera* leaf extract on anti-inflammatory response, oxidative status, and goblet cell response against *Eimeria papillate* infection in mice. Methanol was used as a solvent for phytochemicals. The mice were divided into six groups: The first group was the control. The second group was uninfected and treated with 200 mg/kg of extract to test toxicity, and the third, fourth, fifth, and sixth groups of mice received 1 × 10^3^ sporulated *E. papillate* oocysts. The third group received no treatment. The fourth and fifth groups were treated daily with 100 and 200 mg/kg of *V. vinifera* leaf extract, respectively, while the sixth group received 25 mg/kg of toltrazuril daily via gavage. On day 5 p.i., the animals were sacrificed, and jejunum samples were prepared for analyses of histological sections and oxidative stress. The phytochemical analysis using GC-MS of the extract showed the presence of 12 biologically active compounds. The most effective dose was 200 mg/kg, which significantly decreased the number of parasitic stages in the jejunal sections of the mice. The findings demonstrate that *E. papillate* infection in mice results in significant histopathological changes in the jejunum, including inflammation, epithelial vacuolation, villi loss, and a decrease in goblet cell density. When infected mice received treatment, the histological injury score within the infected jejunum tissue decreased by 63%, and the goblet cell quantity dramatically increased, approaching the control values. Finally, the extract ameliorated the changes in glutathione and malondialdehyde due to *E. papillate* infection. The extract was proven to have anti-inflammatory properties and reduce the number of oocysts. Overall, the findings show that *V. vinifera* leaf extract has significant anticoccidial effects in vivo.

## 1. Introduction

*Eimeria* is a widespread parasite that belongs to the Apicomplexa phylum and causes coccidiosis, which is one of the most frequent intestinal illnesses. Because they are the major agents in the transmission of avian coccidiosis, these protozoan parasites pose a concern to animals and poultry. They result in significant damage to the economy of the livestock industry [1,2]. *Eimeria’s* life cycle has two phases: one outside an infected animal and one within the intestines [3,4]. *Eimeria* rapidly proliferates in host cells and damages their digestive tracts [5]. Disease symptoms include diarrhea ranging from being watery to hemorrhagic from mucoid, along with feed malabsorption and decreased feed conversion efficiency, which causes weight loss and, in severe cases, leads to death [6]. Mice infected with *Eimeria papillate* have infections in their jejunum. This infection severely inflames the gut mucosa [7,8]. Mice are the mammalian model that is utilized most frequently in the fields of biomedical research all over the world [9]. Consequently, *E. papillate* is an excellent model for investigating avian coccidiosis [7].

Several drugs are used to eliminate *Eimeria*, such as drugs belonging to the pyridine group (Coyden), quinolone group (nequinate and decoquinate), ionophore group, halofuginone and quinazoline groups, thiamine antagonists (amprolium), and sulphonamide. But there are a number of issues, such as drug-resistant parasites and a deficiency in safe and effective vaccines, that contribute to the fact that the treatment of *Eimeria* with chemical medications does not succeed effectively and has a wide range of harmful effects [10]. For example, amprolium is a synthetic medication used in anti-coccidiosis therapy. Amprolium hydrochloride (1-[(4-amino-2-propyl-5-pyridinyl) methyl]-2-methyl pyridinium chloride monohydrochloride) is a small synthetic molecule that is a quaternized derivative of pyrimidine (C14H19ClN4). It belongs to a group of organic compounds called methyl pyridines, which have a pyridine ring with a methyl group in two places. This thiamine (vitamin B1) analog interferes with thiamine metabolism by blocking thiamine uptake and, thus, preventing the synthesis of carbohydrates in coccidiosis [11].

Although toltrazuril-based combination therapy against coccidiosis has shown exceptional responses, the limited medicinal options demand novel therapeutics. Furthermore, drug resistance is the cause in most cases, and new medications are proposed to overcome the resistance in addition to conventional therapeutics. The pathogenicity studies showed that the disease burden could be reduced by improving *Eimeria* diagnosis, strengthening prevention, parasitic-correct therapies, and adopting strategies to prevent drug resistance [12]. Despite the advances in treatment strategies, there are still challenges to discovering and developing new therapeutics to improve the quality of life and increase the survival rate [13]. One of the most critical resources for developing highly selective novel pharmaceuticals is using plants for medical purposes [14]. Recent research has found that natural plant sources can be an effective and offer a risk-free method for treating a variety of parasitic illnesses. Drug discovery based on natural products and secondary metabolites is considered an alternative approach for anti-coccidiosis therapy. Herbal medicines have advantages over modern treatments, including cost-effectiveness and affordability, thus encouraging herbal-based drug discovery. Several naturally occurring, semisynthetic, and synthetic anti-coccidiosis medications are on the market [15].

*Vitis vinifera* leaves is often regarded as one of the most significant plant species; it is a traditional therapeutic herb that contains phenolic compounds and is, therefore, a source of bioactive polyphenolic compounds, such as flavonoids and stilbenes [16]. In addition to polyphenolic components, grapes also contain beneficial fatty acids, including unsaturated fatty acids such as linoleic acid and oleic acid, which boost the nutritional value of grape leaf when it is used in food or as a dietary supplement [17]. Research has shown that grapes possess anti-inflammatory, antioxidant, cardio-protective, and anticancer properties. These benefits could be attributed to the presence of linoleic acid, tocopherol, carotenoids, and phytosterols, in addition to some polyphenolic compounds, such as proanthocyanidins, resveratrol, and quercetin [18]. Grapes contain a phytochemical called stilbene resveratrol, which is a powerful antioxidant. There is some evidence that it may help protect against cardiovascular illnesses [17]. In addition, research has indicated that an extract of whole grape berries may be more effective in bringing about health benefits, such as cytotoxic action against cancer cells of the lung, breast, and human liver than an extract of pure phytochemical molecules [19]. *V. vinifera* extracts obtained from various sources have demonstrated antifungal and antibacterial activities, lending credence to the plant’s historical application in various medicinal contexts, such as bacteria, viruses, parasites, and fungi. Wine, winery byproducts, and the bioactive components of these substances are all effective against human pathogens. They are employed as a treatment for pain relief due to the analgesic and anti-inflammatory qualities that they induce in mice infected with *E. papillate* [20,21].

Grape extract, in particular, has been shown to have various pharmacological and therapeutic effects, including antioxidative, anti-inflammatory, and antimicrobial properties [22]. In vivo testing revealed that Astragalus membranaceous possesses substantial anticoccidial activity against E. papillate infection [23]. Study results show that *V. vinifera* has a favorable, inhibitory, and protective effect against the sporulation of coccidian oocysts both in vitro and in vivo in mice [24]. The leaves of *V. vinifera* have been used to treat a wide range of illnesses, including hypertension, diarrhea, hemorrhage, varicose veins, and inflammatory turmoil, and to lower blood sugar levels [25]. Additionally, it has been demonstrated to be hepatoprotective against hepatic DNA damage brought on by acetaminophen [26]. 

The objective of this study is to evaluate the efficacy of an extract from the leaves of *Vitis vinifera* in vivo as a bioindicator of the response of goblet cells, in addition to the ameliorative effect on histopathology and oxidative stress caused by *E. papillate* infection in mice.

## 2. Results

### 2.1. The Phytochemical Analysis of VVE

Table 1 shows the most active chemical compounds of *Vitis vinifera* leaf extracts (VVE). The *V. vinifera* extracts contained 12 compounds; the main compound was oleic acid (211348260); other abundant compounds were linoleic acid (177748941), n-Hexadecanoic acid (53856562), 5-hydroxymethylfurfural (42615989), octadecanoic acid (20765095(5)), β-D-glucopyranose, 1,6-anhydrous (13056399), phytol (11334430), glycidyl oleate (9346792), 4H-pyran-4-one, 2,3-dihydro-3,5-dihydroxy-6-methyl (6451576), 9,12-octadecadienoyl chloride, (Z, Z) (6451576), hexanedioic acid, and mono(2-ethylhexyl) ester (3630858) (Figure 1).

### 2.2. Effect of V. vinifera Extract on Fecal Oocyst Output on Day 5 p.i.

In the infected group, the level of oocyst production in the feces reached its highest point: approximately 93% oocysts per gram of feces. In the groups that were given the *V. vinifera* suspension, the production of oocysts was reduced by 34, 49, 64, and 66% with dosages of 100, 150, and 200 mg/kg of extract and 25 mL/kg of reference medicine, respectively (Figure 2); this represented a significant decrease. It was, therefore, quite clear that the dose of 200 milligrams per kilogram had the most significant potential to inhibit the production of fecal oocysts. As a result, for further research, we solely utilized the dose of 200 mL/kg of *V. vinifera*.

### 2.3. Histopathological Study of the Jejunum after Treatment

Hematoxylin and eosin staining was performed on a section of the mice’s jejunum to conduct additional research on the pathological changes brought on by the infection within the small intestine. Under a light microscope, the histopathological examination revealed considerable structural changes in the intestinal wall. Before receiving the extract treatment, gross pathological abnormalities could be seen in the small intestine in the infected group. These changes included edema, hyperemia, and petechial hemorrhages in the mesenteries.

These changes included bleeding in the intestinal villi, desquamation at the ends of the intestinal villi, and desquamation of mucosal epithelial cells, in addition to the development of parasitic stages (Figure 3). Villus height and crypt depth were significantly reduced in the infected group compared to the uninfected control mice. On the fifth day of treatment, the damage to the intestine caused by *E. papillate* was at its worst. As a direct consequence of this discovery, the tissue of the small intestine was harvested from the rats on the fifth day of therapy and used for subsequent molecular analysis.

### 2.4. Stages of Parasites

Portions of the jejunal villi that have been stained with hematoxylin and eosin are seen in Figure 1. In mice that had been infected with *E. papillate* and given various dosages of *V. vinifera*, the total number of intracellular *E. papillate* stages—including meronts, gamonts, and developing oocysts—was significantly reduced, particularly at a dose of 200 mg/kg of *V. vinifera*. This was especially obvious in the mice that had been given extracts to treat their condition. The mice that were given E. papillate became infected with the disease after receiving the organism. Epithelial cells in the jejunum of mice that had been experimentally infected with *E. papillate* oocysts formed distinct stages of the parasite. These mice were used to study the development of *E. papillate*. (Figure 4). Compared to the infected group, the treatment with *V. vinifera* resulted in a significant reduction of 64% in the number of parasitic stages counted per ten villous–crypt units (*p* ≤ 0.05).

### 2.5. The Effect of V. vinifera Leaf Extract on Goblet Cells in the Jejunum

Through the use of microscopic analysis of Alcian blue-stained jejunal sections, it was discovered that *E. papillate* infection led to a reduction in the number of goblet cells in the jejunum that was statistically significant (*p* ≤ 0.05) when compared to the non-infected group. On the other hand, when compared to the infected group, the jejunum of mice that had been given VVLE showed considerable increase in the number of goblet cells (Figure 5, Figure 6 and Figure 7).

### 2.6. Effect of VVLE on Oxidative Stress in the Jejunum

In the infected control group, infection with *E. papillate* resulted in a highly significant increase in malondialdehyde (*p* ≤ 0.001) and a substantial drop in TP and GSH (*p* ≤ 0.01, *p* ≤ 0.001), compared to the non-infected group. The treatment with a dose of 200 mg/kg of *V. vinifera* leaf extract, on the other hand, generated a significant drop in MD (*p* ≤ 0.001) and significant increases in GSH and TP (*p* ≤ 0.05 and *p* ≤ 0.001, respectively) (Figure 8, Figure 9 and Figure 10, respectively).

## 3. Discussion

It has been demonstrated in a number of studies that *V. vinifera* extract has the potential to be effective in treating a wide range of illnesses and conditions that affect the body [27,28]. It is well established that most anticoccidial treatments will produce a reduction in intracellular *Eimeria* stages and diminish their influence on the rate at which oocysts are shedding. The high concentrations of polyphenolic chemicals present in plant extracts may be responsible for the anticoccidial effects [8,29]. These chemicals have a significant antimicrobial impact because they interfere with the cell walls and membranes of microorganisms, changing the permeability of cell membranes to cations and water. This results in defective membrane functions and cellular component leakage, ultimately leading to cell death [30]. 

According to earlier studies, the carbohydrate content of intestinal epithelial cells and the severe structural abnormalities that result from the consumption of parasitic stages of *Eimeria* are to blame for the loss in cell membrane functions [31]. Because plant extracts contain bioactive compounds that contribute to improving the jejunal histological architecture after treatment, these benefits are realized. An infection with *E. papillate* is associated with a significant inflammatory response and oxidative damage to the jejunum of mice. The administration of *V. vinifera* to infected animals resulted in effective regulation of the oxidative damage that had been caused to the tissue of the mice’s jejunum [32,33]. According to the chronic toxicity studies by Thagfan et al. on mulberry extract, 200 mg/kg is safe without any specific toxicity or side effects [34]. This study tested three doses of *V. vinifera* (100, 150, and 200 mg/kg) as a natural target product against coccidia. It was also discovered that a grape leaf extract dosage could improve mice’s levels of phenols produced by their immune responses and have significant activity against cholinesterase and oxidative stress in a retrograde amnesia mouse model [28,35]. According to the findings of this research, a dose of 200 mg/kg had the highest anticoccidial efficacy out of all the doses that were examined. In addition, we demonstrated that the presence of *V. vinifera* disrupted the life cycle of *E. papillate* at every stage, as well as the oocyst sporulation process. This was evidenced by the considerable decrease in the number of parasitic developmental stages found in the jejunum as well as an increase in the number of fecal oocysts excreted. In addition, a dose-dependent pattern of efficient reduction in the amount of oocyst sporulation was discovered. The results of this research indicated that *V. vinifera* is an effective medication for the treatment of coccidiosis. The anticoccidial activity may be linked to the polyphenol content of the plant, which acts on protozoan development by interacting with cholesterol present on parasitic cell membranes. This results in the suppression and cessation of the life cycle of the parasite and ultimately parasitic death [36].

Bidens pilosa uses a similar mechanism to treat coccidiosis in hens by inhibiting oocyst sporulation and sporozoite invasion into cells [37]. Additionally, a suspension of *V. vinifera* appears to limit oocyst sporulation, which will ultimately lead to a reduction in the propagation of infection [37,38]. It is common knowledge that goblet cells can act as a dynamic defense system against harmful bacteria, viruses, and parasites by modifying the mucus contents and growing in both quantity and size [39]. It has been proven that stem cells that develop into goblet cells are restricted to the crypts of the digestive tract [40]. An analysis of histological sections of the jejunum conducted by Thagfan et al. revealed that the parasitic stages of *E. papillate* were most typically identified in the crypt region [41]. The significant decline in the number of goblet cells seen in the infected group might be explained by the fact that these cells were exposed to stem cells that had been parasitized and had lost their capacity to generate new cells on their own during the course of the infection [37].

The strong potential effect of *V. vinifera* leaf extract is due to the antioxidant and anti-inflammatory [42] activities of components of the plant extract. According to [43], a coccidial infection will produce an imbalance between the body’s innate antioxidant defenses and its generation of free radicals. The findings showed that *E. papillate* infection is associated with oxidative damage to the mice’s jejunum, which causes antioxidant enzymes to be depleted and GSH, TP, and MDA levels to be reduced. These oxidative parameters are crucial for shielding an animal’s body from free radical damage during *Eimeria* infection. Antioxidant enzymes use plant extracts’ active sites to detoxify the accumulation of reactive oxygen species that cause destructive changes in the jejunal epithelium [44,45]. V. vinifera significantly prevents the infection-induced loss of these markers and upregulates their activity, which is typically decreased during oxidative damage induced by infection. The synthesis of several carbonyl compounds, such as MDA, results from the oxidation of lipid peroxides [46]. 

In this work, anticoccidial medicines in the form of herbal extracts were employed to combat *E. papillate* infection. We conducted this study after finding that therapy with V. vinifera disrupts the growth process of this parasite, which, in turn, leads to a rise in the number of goblet cells arising from the harm done. Additionally, Forder and Thagfan previously reported that mulberry extract treatment has a significant effect and activity against *Eimeria* [41,47].

Based on the observations described above, we might conclude that *V. vinifera* possesses potent anticoccidial efficacy. This is supported by the observations showing a decline in the release of oocysts and sporulation, a reduction in the number of stages of parasite development found in the jejunum, and a recovery of normal goblet cell numbers.

## 4. Materials and Methods

### 4.1. Examination of experimental mice

The feces of the mice were analyzed six days before the beginning of the experiment using the flotation method and microscopic inspection of fallen oocysts in the feces, as described by Ryley et al. (1976). This was performed to guarantee that the animals did not have any coccidia [48].

### 4.2. Ethical Approval

This research was carried out according to the ethical guidelines for the use of animals established by the Kingdom of Saudi Arabia (Ethics Committee of King Saud University, approval number: KSU-SE-21-86).

### 4.3. Preparation of Vitis Vinifera Leaf Extracts

The *V. vinifera* leaf extracts were prepared using grape leaves from farms in Riyadh, Saudi Arabia. The botanical identity of the plant was confirmed by a taxonomist at the Department of Botany, King Saud University. The leaves were air-dried at 42 °C, ground into a powder, and then extracted with 80% methanol for 24 h at +4 °C [49]. The resulting extract was concentrated and dried in a rotating vacuum evaporator (Yamato RE300, Tokyo, Japan). The powder was dissolved in distilled water for the investigations of various experiments.

### 4.4. Phytochemical Analysis

Kanthal et al.’s procedure was used to conduct the phytochemical analysis of VVLE (2014). Gas chromatography–mass spectrometry (GC-MS) was performed using a 7000D Triple Quad GC-MS equipment (Agilent Technologies, Thermo Scientific, Austin, TX, USA). Thermo Scientific Trace GC Ultra and an ISQ single quadruple MS were used (Miami, CA, USA). The Agilent 7890A and 5975C inert XL EI/CI MSDs have a single quadrupole mass analyzer. A capillary HP-5MS UI (Ultra Inert) was used, with a length of 25 m, an inner diameter of 0.25 mm, a film thickness of 0.25 m, a stationary phase with 5% phenyl, and methylpolysiloxane (low polar). Helium was the carrier gas; the flow rate was 1 mL/min, the split ratio was 50:50, and the temperature was 250 °C. The GC-MS was run at 30 °C (5 min), at a rate of 5 Co/min, and then the temperature was increased to 250 °C (40 min). The temperature of the transfer line was 250 °C. The ionization energy in the MS environments was 70 eV. Full scan detection mode was used, with a mass range of 50–500 Da. The solvent delay was 1 min. Methanol was the sample solvent. The Wiley 9 database, responses, and libraries were used to identify the compounds. The components of the test substance were determined by name, molecular weight, molecular formula, and peak area. 

### 4.5. Mice

The experiment was performed on mature male C57BL/6 mice that ranged in age from 10 to 13 weeks and weighed an average of 21 g per mouse. The animals were maintained in pathogen-free conditions in accordance with the procedures that were stated. The environment had a controlled temperature of 21 °C and a 50:50 light and darkness regimen, with standard diet and water provided ad libitum.

### 4.6. Sporulation of Oocysts

Professor Al-Quraishy obtained *E. papillate* oocysts in the parasitology laboratory, Zoology Department, in the College of Science at King Saud University in Saudi Arabia via passage through coccidian-free mice to stimulate the virulence of oocysts before they were used to infect the hosts, where they were maintained. Several mice were infected with an initial stock of sporulated oocysts. The mice had their feces collected to obtain unsporulated oocysts five days after infection, and new oocysts were sporulated so they could be used in the experiment. Fecal samples were collected using a modified version of the McMaster technique. For oocyst floatation, the fecal pellets of each mouse were suspended in a solution containing 2.5% potassium dichromate (w/v) in saturated sodium chloride (NaCl). The solution was then removed from the pellets by washing. After adjusting the total number of newly collected oocysts, one thousand sporulated oocysts suspended in one hundred microliters of water were orally gavaged into each mouse. This was performed to keep track of the oocyst-shedding numbers [50].

### 4.7. Treatment Design

The mice were distributed into seven groups, featuring five in each group. The first control group received 100 µL of sodium chloride (0.9% NaCl), and the second group received 200 mg/kg of *V. vinifera* leaf extract only, without infection, to determine toxicity. For the third, fourth, fifth, sixth, and seventh groups, the mice received 1 × 10^3^ sporulated E. papillate oocysts. The third group did not receive any treatment. The fourth, fifth, and sixth groups were treated daily with 100, 150, and 200 mg/kg of *V. vinifera* leaf extract, respectively, while the seventh group received 25 mg/kg of toltrazuril every day for five days via gavage.

### 4.8. Sample Collection and Number of Oocysts in the Jejunum

On day 5, each mouse was sacrificed, and pieces of the jejunum were collected for histological examination and oxidative stress marker analysis; they were fixed in 10% neutral formalin buffer, dehydrated in ethanol, embedded in paraffin wax, and cut into 5 μm thick sections. Hematoxylin and eosin (H&E) staining was used on the sections (DRURY & WALLINGTON, 1980). The fixed tissues were then processed to assess parasitic developmental stages (meronts, gamonts, and developing oocysts in the groups that had been infected and treated with the plant extract or toltrazuril) based on the number of oocysts in every 10 well-oriented villous–crypt units (VCU) in all animals, utilizing an Olympus BX61 light microscopy (Tokyo, Japan).

### 4.9. Histopathological Examination

Excisions were made in the small intestine, and the samples were immediately washed in physiological saline and then fixed in phosphate-buffered formalin at a 70% concentration. After being set with formalin, the samples were encased in paraffin and then sectioned longitudinally at a thickness of 5 μm. Hematoxylin and eosin were used for staining after the segment of jejunum had been deparaffinized and dehydrated (Sigma). Using an Olympus BH2 microscope, observations of the microstructures of the small intestine were carried out (DP71, Olympus, Tokyo, Japan). Using an Olympus Image Analysis System, the height of the villus and the depth of the crypt of each of the ten well-oriented villi of each mouse were measured and recorded (Olympus 6.0, Tokyo, Japan).

### 4.10. Counting of Goblet Cells

The paraffin slices were rehydrated gradually in declining concentrations of ethanol and water before being deparaffinized with xylene. The sections were stained with Alcian blue (Sigma) to determine the number of goblet cells. Each animal’s jejunum was counted for the number of goblet cells in at least ten well-oriented villous–crypt units (VCUs). The average number of goblet cells per ten villi was calculated to represent the results [51].

### 4.11. Oxidative Stress

After weighing the jejunum from each of the different groups of mice, it was immediately homogenized in phosphate-buffered saline (PBS) in a buffer solution that had a pH of more than 7.4, and it was centrifuged at 5000× *g* for 15 min at a temperature of 4 degrees Celsius. After collecting the supernatant, 10 percent of the volume was utilized to determine the levels of oxidative stress markers. The levels of GSH and MDA in the jejunum were measured according to [52,53], respectively. In addition, total protein activity was measured in the mouse jejunum using the methodology described in [52,54].

### 4.12. Statistical Evaluation

For statistical comparisons between two groups, Student’s *t*-test was used. For statistical comparisons between more than two groups, an analysis of variance (ANOVA) test was used with post hoc Tukey’s test. The software program SPSS 12.0 was used to carry out the statistical analyses, and graphing was accomplished using version 5.0 of the program Graphpad Prism, Origin 2018 (Systat Software, Inc., Chicago, IL, USA). All findings were reported as mean accompanied by the standard error of the norm, with the threshold for statistical significance set at *p* ≤ 0.05.

## 5. Conclusions

Reducing parasites is essential for organisms’ health, which can contribute to sustainable production that keeps consumers healthy. Grape leaf extract has been shown to have anti-inflammatory characteristics. It is deduced that *V. vinifera* leaf extracts are strong against *E. papillate*-induced infection activity in light of the results mentioned above. This is shown by a decrease in the production of oocysts and sporulation, a reduction in parasitic developmental stages in the jejunum, and the recovery of normal goblet cell counts. There is significant improvement in the histological changes of the mice’s jejunum and antioxidant status. More research is required to gain an understanding of the mechanism by which grape leaf extract affects both the parasite and the host mice. To learn more about the plant’s pharmacological and therapeutic properties, more experimental studies and clinical research are required. This will help in the development of important therapeutic drugs based on the active phytochemical components of *V. vinifera*.

## Figures and Tables

**Figure 1 pharmaceuticals-16-00928-f001:**
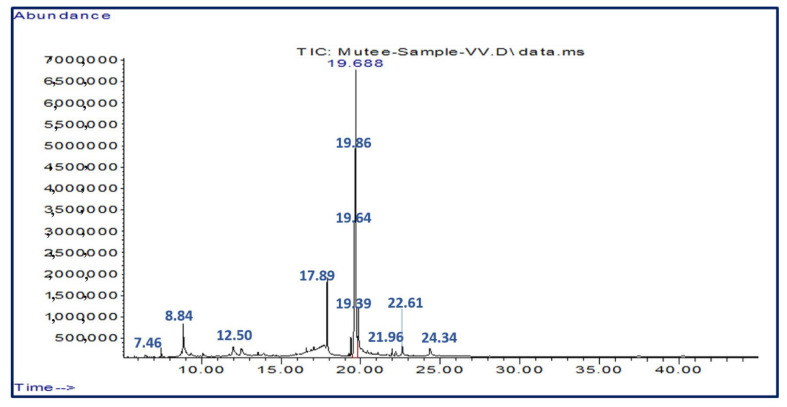
Infrared spectroscopy of *V. vinifera* leaf extracts.

**Figure 2 pharmaceuticals-16-00928-f002:**
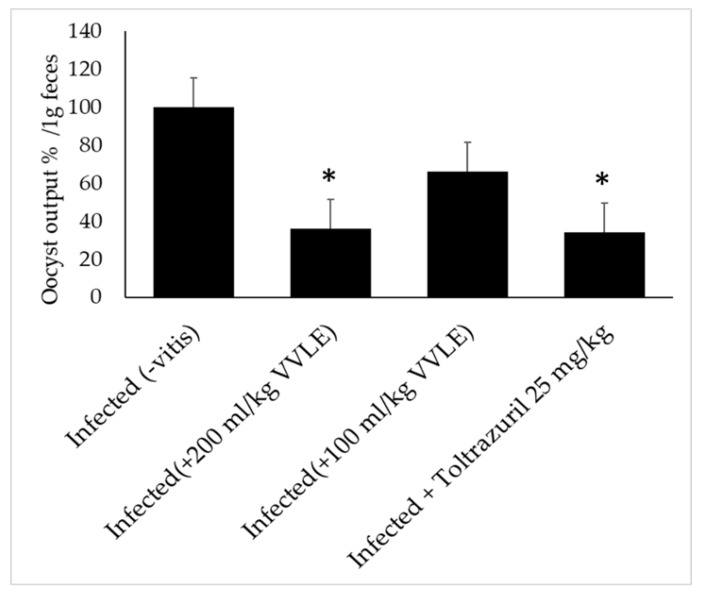
Effect of *V. vinifera* leaf extracts: oocyst production decrease in parasitic stages in mice with *E. papillate* infection in the jejunum. All values are displayed as mean ± SE. * Significant deviation from the control (*p* ≤ 0.05).

**Figure 3 pharmaceuticals-16-00928-f003:**
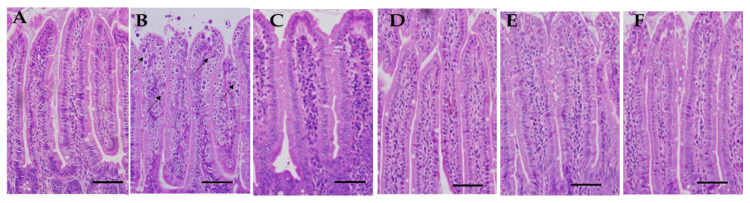
Histological changes in the jejunal sections of mice due to infection with *Eimeria papillate* sporulated oocysts that have been stained with hematoxylin and eosin on day 5 p.i.: (**A**) control (-VVLE); (**B**) infected (-VVLE) indicated by arrows; (**C**) non-infected (+VVLE); (**D**) 100% (infected + VVLE); (**E**) 200% (infected + VVLE); and (**F**) infected + 25 mg/kg of toltrazuril. Scale bar = 50 μm.

**Figure 4 pharmaceuticals-16-00928-f004:**
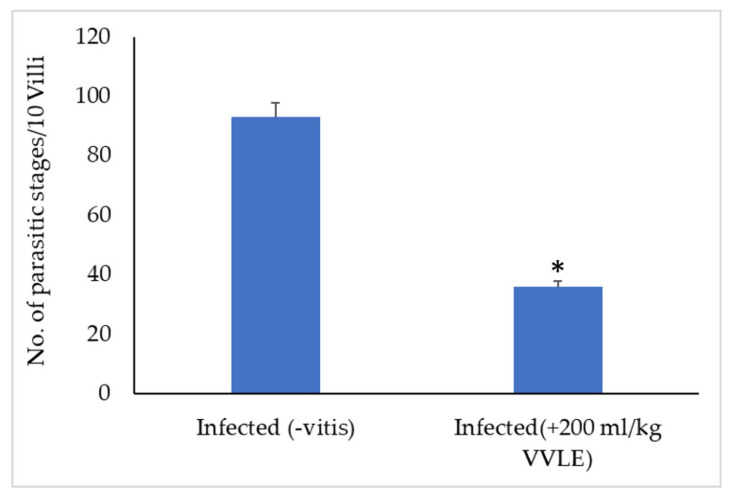
Oocyst production patterns of mice treated with varying dosages of *V. vinifera* leaf extracts on day 5 post-treatment and infection with *E. papillate* (all values reported as mean ± SE). * Significant deviation from the control group (*p* ≤ 0.05).

**Figure 5 pharmaceuticals-16-00928-f005:**
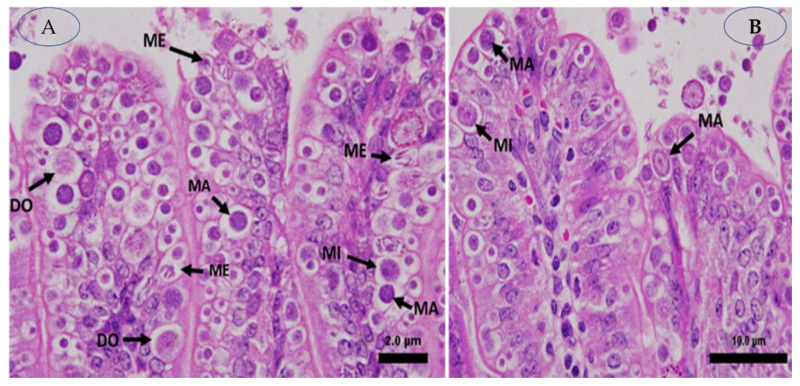
On day 5 post-inoculation, *E. papillate* infection was found in the jejunum: (**A**,**B**) distinct developmental phases. Meront (ME), macrogamont (MA), and microgamont (MI) shown in (**A**). Macrogamont (MA), microgamont (MI), and developing oocyst (DO) shown in (**B**). Hematoxylin and eosin, also known as H&E, were used to stain the sections.

**Figure 6 pharmaceuticals-16-00928-f006:**
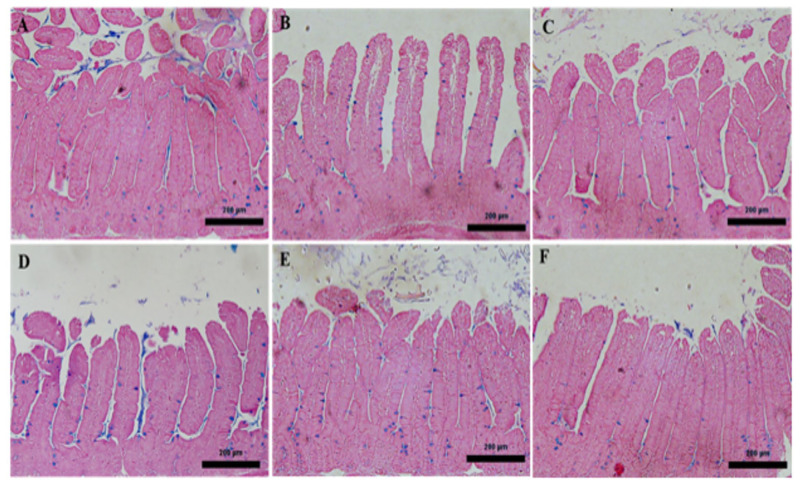
The influence of extracts from the leaves of *V. vinifera* on the number of goblet cells that are present in the jejunum of mice that have an infection caused by *E. papillate*. (**A**) represents the control group; (**B**) represents the infected group that has developmental stages; (**C**) represents the non-infected *V. vinifera* leaf extract group; (**D**) represents the treated group that received 100 mg/kg of *V. vinifera* extract; (**E**) represents the treated group that received 200 mg/kg and has fewer goblet cells; and (**F**) represents the infected group that underwent therapy with toltrazuril. Alcian blue was used to stain the sections. Scale bar = 50 µm.

**Figure 7 pharmaceuticals-16-00928-f007:**
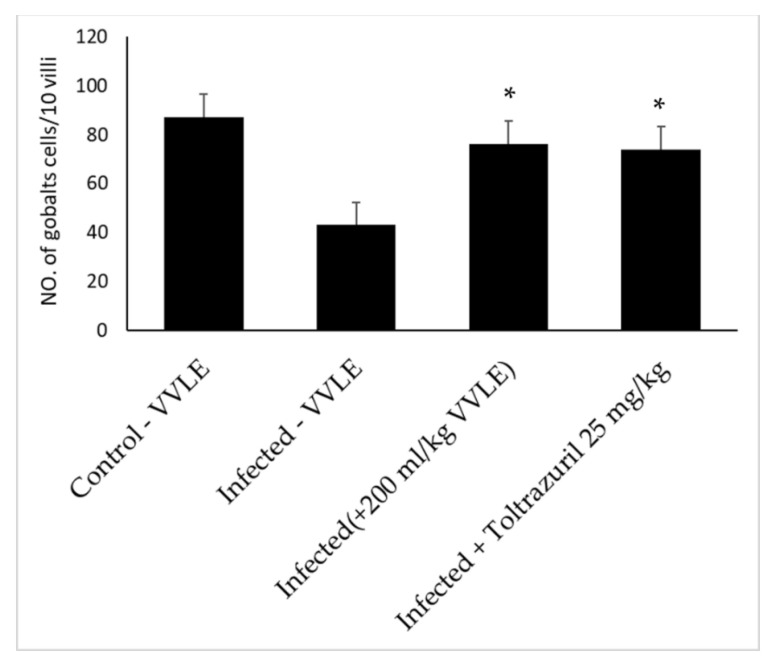
Effect of *V. vinifera* leaf extracts: reduction in goblet cell population in *E. papillate*-infected mouse jejunum. All values are expressed as mean ± SE. Significant deviation from the control group (*p* ≤ 0.05) is denoted with *.

**Figure 8 pharmaceuticals-16-00928-f008:**
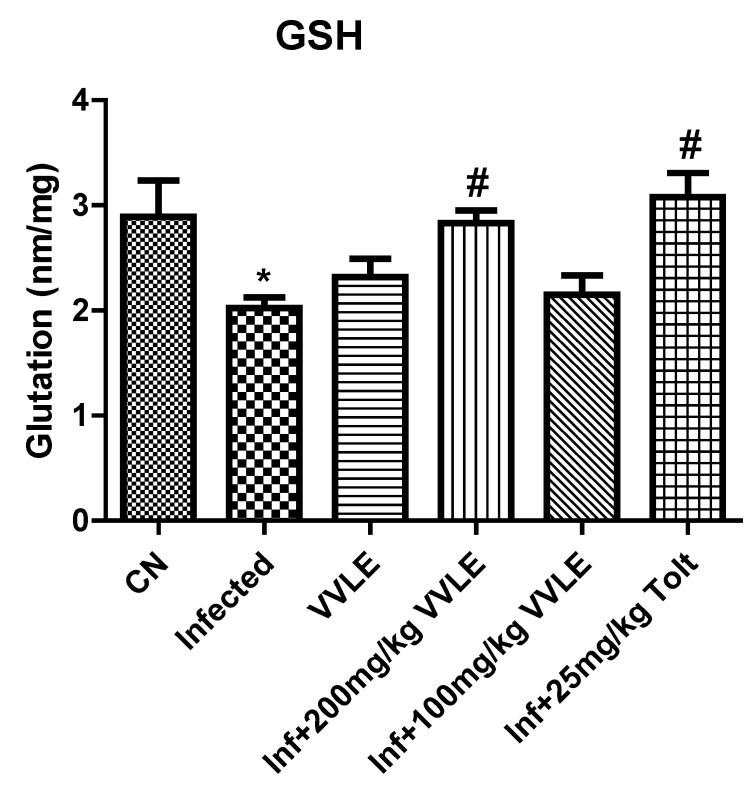
The effect of *V. vinifera* leaf extracts on the levels of oxidative stress markers (GSH) in the jejunum of mice infected with *E. papillate*. All values are shown as mean ± SE. CN: control; Inf: infected; VVLE: *Vitis vinifera* leaf extract. * Significant difference when compared to the control group (*p* < 0.01). # Significant difference when compared to the control group (*p* < 0.05).

**Figure 9 pharmaceuticals-16-00928-f009:**
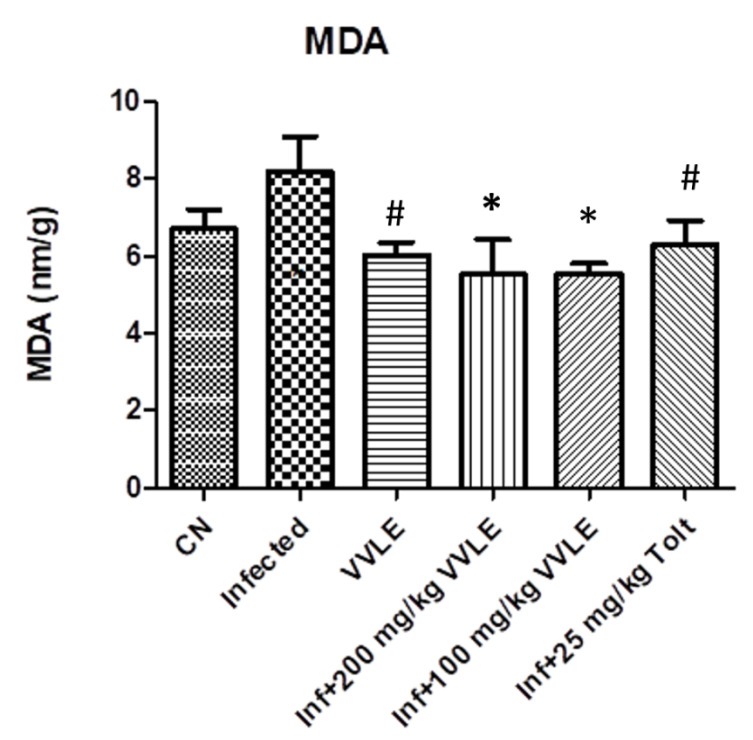
The effect of VVLE on the levels of oxidative stress markers (MDA) in the jejunum of mice infected with *E. papillate*. All values are shown as mean ± SE. CN: control; Inf: infected; VVLE: *Vitis vinifera* leaf extract. * Significant difference when compared to the control group (*p* < 0.05). # Significant difference when compared to the control group (*p* < 0.05).

**Figure 10 pharmaceuticals-16-00928-f010:**
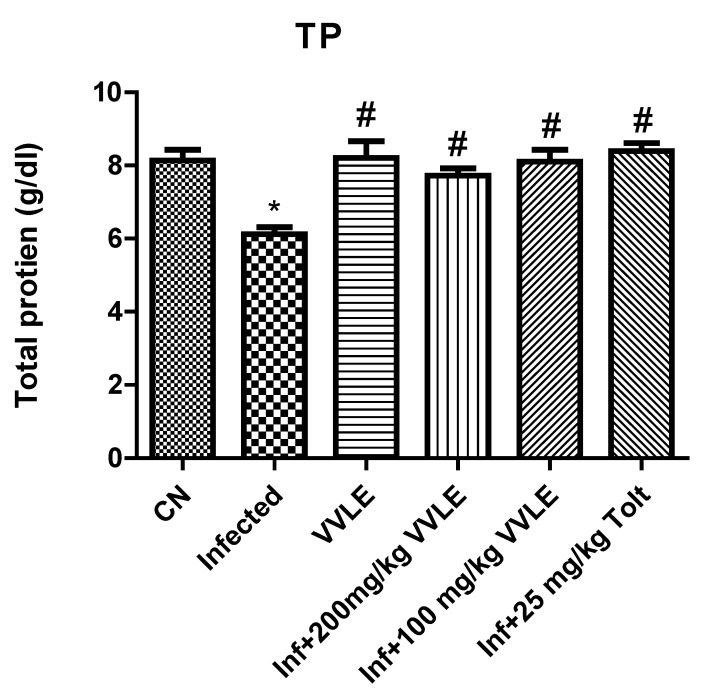
The effect of *V. vinifera* leaf extracts on the levels of oxidative stress markers (TP) in the jejunum of mice infected with *E. papillate.* All values are shown as mean ± SE. CN: control; Inf: infected; VVLE: *Vitis vinifera* leaf extract. * Significant difference when compared to the control group (*p* < 0.05). # Significant difference when compared to the control group (*p* < 0.05).

**Table 1 pharmaceuticals-16-00928-t001:** Phytochemical compounds identified through GC-MS in *Vitis vinifera* leaf extracts.

Retention Time (min)	Phytochemicals	Chemical Group	Molecular Weight	Formula	Peak Area%
7.46	4H-Pyran-4-one, 2,3-dihydro-3,5-dihydroxy-6-methyl-	Flavonoid	144	C6H8O4	1.16
8.84	5-Hydroxymethylfurfural	Furanic aldehyde	126	C6H6O3	7.66
12.50	β-D-Glucopyranose, 1,6-anhydro-	Pyranose	162	C6H10O5	2.35
17.89	n-Hexadecanoic acid	Fatty acid	256	C16H32O2	9.68
19.39	Phytol	Diterpene	296	C20H40O	2.04
19.64	Linoleic acid	Polyunsaturated fatty acids (PUFA)	280	C18H32O2	31.93
19.69	Oleic acid	Monounsaturated fatty acids (MUFA)	282	C18H34O2	37.97
19.86	Octadecanoic acid	Saturated fatty acids	284	C18H36O2	3.73
21.96	Hexanedioic acid, mono(2-ethylhexyl) ester	Adipic acid ester	258	C14H26O4	0.65
22.61	Glycidyl oleate	Glycidyl ester	338	C21H38O3	1.68
24.34	9,12-Octadecadienoyl chloride, (Z, Z)-	Fatty acyl chlorides	298	C18H31ClO	1.15

## Data Availability

Data is contained within the article.

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
