# Peer review of "Pharmacological Effects of Grape Leaf Extract Reduce Eimeriosis-Induced Inflammation, Oxidative Status Change, and Goblet Cell Response in the Jejunum of Mice"

_pharmaceuticals, 2023, doi:10.3390/ph16070928_

Round 1

Reviewer 1 Report (Previous Reviewer 1)

Please double check for any other possible spelling errors.

Author Response

Reviewer 1

Please double-check for any other possible spelling errors.

Done check spelling errors

Reviewer 2 Report (Previous Reviewer 2)

The study design is great. But I have some comments:

1. Add kindly more updated references to the Introduction part (2021-2023).

2.  As mentioned in the following paper "https://doi.org/10.3390/medicines4040089" Vitis vinifera have verious pharmacological effects, use this paper to enrich your introduction part.

3. After mentioning the plant full name (Vitis vinifera) you can then write its name in abbreviation manner (V. vinifera).

4. All the Latin words throughout the whole manuscript must be in Italic manner.

5. No need to use VVLE to be your work clear for readers.

6. In section ‘Preparation of Viti’s vinifera leaf extracts” add how you identified the plant species and who is the taxonomist who classified the species and where the herbarium deposited.

7. The GC- Mass not a suitable technique used to identify the chemical constituents of Vitis vinifera leaves extracts kindly replace this test with LC-MS-MS.

8. The whole manuscript needs major grammar, typos and editing correction by a native speaker specialist in biological and biomedical sciences.

The whole manuscript needs major grammar, typos and editing correction by a native speaker specialist in biological and biomedical sciences.

Author Response

Comments and Suggestions for Reviewer 2

The study design is great. But I have some comments:

  1. Add kindly more updated references to the Introduction part (2021-2023).

Done

  1. As mentioned in the following paper "https://doi.org/10.3390/medicines4040089" Vitis vinifera have verious pharmacological effects, use this paper to enrich your introduction part.

One of the most important plants is Vitis vinifera leaf, a traditional medicinal plant that contains phenolic compounds and is, therefore, a source of bioactive polyphenolic compounds such as flavonoids and stilbenes (Leifert and Abeywardena, 2008). In addition to polyphenolic components, grapes also contain beneficial fatty acids, including unsaturated fatty acids such as linoleic acid and oleic acid, which boost the nutritional value of grape leaf when it is used in food or as a dietary supplement (Sabra et al., 2021). Research has shown that grapes possess anti-inflammatory, antioxidant, cardio-protective, and anticancer properties. These benefits could be attributed to the presence of linoleic acid, tocopherol, carotenoids, and phytosterols, in addition to some polyphenolic compounds such as proanthocyanidins, resveratrol, and quercetin (Garavaglia et al., 2016). Grapes contain a phytochemical called stilbene resveratrol, which is a powerful antioxidant. There is some evidence that it may help protect against cardiovascular illnesses (Sabra et al., 2021). In addition, research has indicated that an extract of whole grape berries may be more effective in bringing about health benefits, such as cytotoxic action against cancer cells of the lung, breast, and human liver, than an extract of a pure phytochemical molecule (Balasubramani et al., 2019).

  1. After mentioning the plant full name (Vitis vinifera) you can then write its name in abbreviation manner (V. vinifera).

Done

  1. All the Latin words throughout the whole manuscript must be in Italic manner.

Done

  1. No need to use VVLE to be your work clear for readers.

Done

  1. In section ‘Preparation of Viti’s vinifera leaf extracts” add how you identified the plant species and who is the taxonomist who classified the species and where the herbarium deposited.

The botanical identity of the plant was confirmed by a taxonomist at the Department of Botany, University of King Saud.

  1. The GC- Mass not a suitable technique used to identify the chemical constituents of Vitis vinifera leaves extracts kindly replace this test with LC-MS-MS.

We examined the plant with the available capabilities (GC-MS) and (IR)

The authors appreciate the suggestion of the expert reviewer but we are confident that GC-MS analysis is a reliable technique to analysis any plant extract. The analysis was aimed to reveal the major components of the crude extract that were sufficient for our research interest in future. The GC-MS has been widely used for analysis and a great number of contemporary investigators still use the technique to report phyto-analysis (Konappa et al., 2020; Gomathi et al., 2015; Akubugwo et al., 2022). Our So, we did not feel to do LC-MS of the extract in the present study. However, we shall keep the suggestion in mind in our future plant based extracts analysis.

The findings obtained by doing infrared spectroscopy on samples of VVLE. The information was gathered with the assistance of an FT-IR spectrometer, and the results were found to fall somewhere in the region of 400 to 4000 /cm-1.

Absorption(cm-1)

Appearance

Transmittance (%)

Group

Compound class

3425.53

medium

12

N-H stretching

Aliphatic primary amine

2093.10

strong

47

N=C=S stretchy

Isothiocyanate

1641.41

strong

25

C=C stretching

Alkene

1209.12

strong

37

C-O stretching tertiary

alcohol

1045.64

Strong broad

35

CO-O-CO stretching

Anhydride

410.42

strong

3

C-H bending

1,2-disubstituted

References:

Konappa, N., Udayashankar, A.C., Krishnamurthy, S. et al. GC–MS analysis of phytoconstituents from Amomum nilgiricum and molecular docking interactions of bioactive serverogenin acetate with target proteins. Sci Rep 10, 16438 (2020). https://doi.org/10.1038/s41598-020-73442-0

Gomathi D, Kalaiselvi M, Ravikumar G, Devaki K, Uma C. GC-MS analysis of bioactive compounds from the whole plant ethanolic extract of Evolvulus alsinoides (L.) L. J Food Sci Technol. 2015 Feb;52(2):1212-7. doi: 10.1007/s13197-013-1105-9.

Akubugwo E I , Okezie Emmanuel et al., GC-MS Analysis of the Phytochemical Constituents, Safety Assessment, Wound Healing and Anti-Inflammatory Activities of Cucurbita pepo Leaf Extract in Rats. Sci. Pharm. 2022, 90, 64. https://doi.org/ 10.3390/scipharm90040064

  1. The whole manuscript needs major grammar, typos, and editing correction by a native speaker specialist in biological and biomedical sciences.

A native English speaker has edited the revised manuscript language.

Reviewer 3 Report (Previous Reviewer 4)

With respect to the authors perception, in the reviewer opinion, the changes made to the manuscript trying to clarify “the real contribution of the obtained results to the development of veterinary sciences, and especially the prophylaxis/treatment of coccidiosis in any food producing animal“ beside the “scientifically documentation of the pathogenicity of Eimeria papillate for any food producing animals” are minor and, in principle, it does not differ from the original version.

After a careful second review of the manuscript's scientific merit, I believe that the present paper was don't corrected according to the reviewer's recommendations. Therefore, I cannot recommend this paper in present version to published in Pharmaceuticals scientific journal.

 In addition, the authors did not manage to overcome the barrier of italic writing of the species scientific names (e.g. 347, 413, 439). In some parts, the reference list is not in agreement with the journal requirement.

Author Response

Comments and Suggestions for Authors

With respect to the authors perception, in the reviewer opinion, the changes made to the manuscript trying to clarify “the real contribution of the obtained results to the development of veterinary sciences, and especially the prophylaxis/treatment of coccidiosis in any food producing animal“ beside the “scientifically documentation of the pathogenicity of Eimeria papillate for any food producing animals” are minor and, in principle, it does not differ from the original version.

Mice are used as an experimental model for the application of plant extracts when infected with their own species, such as E. papillate, for easy access and low costs. What applies to this type of Eimeria also applies to other species that affect poultry, rabbits, cattle, sheep, camels, etc. There is no objection to working with the proposals, especially after making sure through the use of a model or hypothetical experiment that the plant extracts of other animals, such as poultry when infected with Eimeria in poultry and Eimeria in poultry rabbits.

After a careful second review of the manuscript's scientific merit, I believe that the present paper was don't corrected according to the reviewer's recommendations. Therefore, I cannot recommend this paper in present version to published in Pharmaceuticals scientific journal.

Work has been done to improve the manuscript in the introduction and the presentation of the results as suggested by the reviewers.

 In addition, the authors did not manage to overcome the barrier of italic writing of the species scientific names (e.g. 347, 413, 439). In some parts, the reference list is not in agreement with the journal requirement.

Correction was done step by step according to the reviewers suggestions.

Round 2

Reviewer 2 Report (Previous Reviewer 2)

The authors did all the required corrections

Minor editing and language corrections 

Reviewer 3 Report (Previous Reviewer 4)

Much improved!

This manuscript is a resubmission of an earlier submission. The following is a list of the peer review reports and author responses from that submission.

Round 1

Reviewer 1 Report

1. Make sure that all scientific names (i.e. "E. papilate", "Vitis vinifera", "V. vinifera", etc.) are italicised throughout the manuscript, including in the Figure descriptions and References.

2. Table 1: 

a. Remove the name "bioactive" from "bioactive phytochemicals" since the research did not perform any bioassay specific for each of the said phytochemicals. The name should simply be "Phytochemicals".

b. Include an additional column after "Phytochemicals" and indicate the chemical family or group where each of the said compounds belong to.

3. Figure 5: lacks the labels A and B per picture and the description is not clear what A and B specifically describes for.

4. Figure 6: The "A control group" is better written as "A control group -VVLE" or "A untreated control group". What is CPLE? Is this supposed to be VVLE? What are pictures E and F?

5. Figure 8: The Y axis should be correctly spelled as "Glutathione". Please improve the labeling of the each bar in the X axis. Please observe consistency. For example, if "Infected +VVLE" is used, the other should be "Infected -VVLE" and not "VVLE-infected", this is confusing. Also 200 mg/mL VVLE and 100 mg/ml VVLE are both infected?? How about with 25 mg/mL Tortrazuril? Please improve and make everything clear. Same labeling should apply for Figures 9 and 10.

6. The Discussion is lacking on the part where the possible roles of each phytochemical in Table 1 might have played in the outcome of the experiment. This should be included with references.

The English language needs to be carefully checked again. There are a few truncated sentences and minor spelling corrections.

Author Response

Review report 1

  1. Make sure that all scientific names (i.e. "E. papilate", "Vitis vinifera", "V. vinifera", etc.) are italicised throughout the manuscript, including in the Figure descriptions and References.

Done

  1. Table 1: 
  2. Remove the name "bioactive" from "bioactive phytochemicals" since the research did not perform any bioassay specific for each of the said phytochemicals. The name should simply be "Phytochemicals".

Done

  1. Include an additional column after "Phytochemicals" and indicate the chemical family or group where each of the said compounds belong to.

Done

  1. Figure 5: lacks the labels A and B per picture and the description is not clear what A and B specifically describes for.

Done

  1. Figure 6: The "A control group" is better written as "A control group -VVLE" or "A untreated control group". What is CPLE? Is this supposed to be VVLE? What are pictures E and F?

Done

  1. Figure 8: The Y axis should be correctly spelled as "Glutathione". Please improve the labeling of the each bar in the X axis. Please observe consistency. For example, if "Infected +VVLE" is used, the other should be "Infected -VVLE" and not "VVLE-infected", this is confusing. Also 200 mg/mL VVLE and 100 mg/ml VVLE are both infected?? How about with 25 mg/mL Tortrazuril? Please improve and make everything clear. Same labeling should apply for Figures 9 and 10.

Done

  1. The Discussion is lacking on the part where the possible roles of each phytochemical in Table 1 might have played in the outcome of the experiment. This should be included with references.

Done

Reviewer 2 Report

1.     Add in abstract part, the type of extract and why you used toxic methanol as solvent.

2.     How you adjusted 200 mg/kg dose?

3.     All the Latin names throughout the whole manuscript must be in Italic manner.

4.     The chemical components of VVmethanolic extract must be identified after fractionation using LC-MS-MS technique and it not appropriate  to analyse them using GC-MS.

5.     In the result section of the identified chemical components as the VV extract contain fatty acids which is not correct as VVE. The leaves mainly contain  five flavonols (quercetin 3-O-glucoside, quercetin 3-O-glucuronide, kaempferol 3-O-glucoside, hyperoside, and rutin), two anthocyanosides (delphinidin 3-O-glucoside and cyanidin 3-O-glucoside), and caftaric acid. Quercetin-3-O-glucuronide, quercetin-3-O-glucoside, and caftaric acid were, in order, the most abundant phenols in the extract; in particular, the quercetin-3-O-glucuronide and the quercetin-3-O-glucoside

6.     Add the voucher specimen code to the plant in section (Preparation of Vitis vinifera leaf extracts).

7.     Add subsection title for paragraph 268-271.

8.     After mentioning the plant or microbe full name you can use its abbreviated name for example Vitis vinifera to be V. vinifera.

9.     The whole manuscript needs major grammar, typo and editing corrections by a native speakers.

The whole manuscript needs major grammar, typo and editing corrections by a native speakers.

Author Response

Review report 2

Comments and Suggestions for Authors

  1. Add in the abstract part, the type of extract and why you used toxic methanol as solvent.

Done on abstract

Methanol is a polar solvent it can readily dissolve polar organics, the best suitable solvent to extract phytochemicals. So, it's giving the highest extraction yields and the highest content of phenolic, flavonoid, alkaloid, and terpenoids.

  1. How you adjusted 200 mg/kg dose?

Because it was the dose that eliminated the parasite without side effects, this was confirmed by the oxidative stress results

  1. All the Latin names throughout the whole manuscript must be in Italic manner.

Done

  1. The chemical components of VVmethanolic extract must be identified after fractionation using LC-MS-MS technique and it not appropriate to analyse them using GC-MS.

The authors appreciate the suggestion of the expert reviewer but we are confident that GC-MS analysis is a reliable technique to analysis any plant extract. The analysis was aimed to reveal the major components of the crude extract that were sufficient for our research interest in future. The GC-MS has been widely used for analysis and a great number of contemporary investigators still use the technique to report phyto-analysis (Konappa et al., 2020; Gomathi et al., 2015; Akubugwo et al., 2022). Our So, we did not feel to do LC-MS of the extract in the present study. However, we shall keep the suggestion in mind in our future plant based extracts analysis.

References:

Konappa, N., Udayashankar, A.C., Krishnamurthy, S. et al. GC–MS analysis of phytoconstituents from Amomum nilgiricum and molecular docking interactions of bioactive serverogenin acetate with target proteins. Sci Rep 10, 16438 (2020). https://doi.org/10.1038/s41598-020-73442-0

Gomathi D, Kalaiselvi M, Ravikumar G, Devaki K, Uma C. GC-MS analysis of bioactive compounds from the whole plant ethanolic extract of Evolvulus alsinoides (L.) L. J Food Sci Technol. 2015 Feb;52(2):1212-7. doi: 10.1007/s13197-013-1105-9.

Akubugwo E I , Okezie Emmanuel et al., GC-MS Analysis of the Phytochemical Constituents, Safety Assessment, Wound Healing and Anti-Inflammatory Activities of Cucurbita pepo Leaf Extract in Rats. Sci. Pharm. 2022, 90, 64. https://doi.org/ 10.3390/scipharm90040064

  1. In the result section of the identified chemical components as the VV extract contain fatty acids which is not correct as VVE. The leaves mainly contain five flavonols (quercetin 3-O-glucoside, quercetin 3-O-glucuronide, kaempferol 3-O-glucoside, hyperoside, and rutin), two anthocyanosides (delphinidin 3-O-glucoside and cyanidin 3-O-glucoside), and caftaric acid. Quercetin-3-O-glucuronide, quercetin-3-O-glucoside, and caftaric acid were, in order, the most abundant phenols in the extract; in particular, the quercetin-3-O-glucuronide and the quercetin-3-O-glucoside

It was added as a column in Table 1 to show the chemical group or family to which the compounds obtained by phytochemical analysis belong. 

Retention time(min)

Phytochemicals

Chemical group

Molecular

weight

Formula

Peak area%

7.46

4H-Pyran-4-one, 2,3-dihydro-3,5-dihydroxy-6-methyl-

flavonoid

144

C6H8O4

1.16

8.84

5-Hydroxymethylfurfural

  Furanic aldehyde

126

C6H6O3

7.66

12.50

β-D-Glucopyranose, 1,6-anhydro-

pyranose

162

C6H10O5

2.35

17.89

n-Hexadecanoic acid

Fatty acid

256

C16H32O2

9.68

19.39

Phytol

diterpene

296

C20H40O

2.04

19.64

Linoleic acid

polyunsaturated Fatty acids (PUFA)

280

C18H32O2

31.93

19.69

Oleic Acid

Monounsaturated fatty acids (MUFA)

282

C18H34O2

37.97

19.86

Octadecanoic acid

Saturated fatty acids

284

C18H36O2

3.73

21.96

Hexanedioic acid, mono(2-ethylhexyl) ester

Adipic acid ester

258

C14H26O4

0.65

22.61

Glycidyl oleate

Glycidyl ester

338

C21H38O3

1.68

24.34

9,12-Octadecadienoyl chloride, (Z, Z)-

Fatty acyl chlorides

298

C18H31ClO

1.15

  1. Add the voucher specimen code to the plant in section (Preparation of Vitis vinifera leaf extracts).

Done

  1. Add subsection title for paragraph 268-271.

Done

  1. After mentioning the plant or microbe full name you can use its abbreviated name for example Vitis viniferato be V. vinifera.

 Done

Reviewer 3 Report

The article describes the evaluation of Vitis vinifera leaf extract (VVLE) on Eimeria papillate infection in mice. The study assessed the effects of VVLE on inflammation, oxidative status, and goblet cell response. The experiments were conducted on mice which were divided into different groups, including a control group and groups receiving varying doses of VVLE or toltrazuril. Histological sections and oxidative stress analysis were performed on jejunum samples. Phytochemical analysis identified 12 biologically active compounds in VVLE. The most effective dose was found to be 200 mg/kg, significantly reducing parasitic stages in the jejunum. E. papillate infection in mice caused histopathological changes, inflammation, and a decrease in goblet cell density. However, treatment with VVLE led to a decrease in histological injury score, an increase in goblet cell quantity, and improvements in glutathione and malondialdehyde levels. The findings demonstrate the anti-inflammatory and anticoccidial effects of VVLE in vivo.

The article is well-written and addresses important issues that should be further developed in future research. However, I have few issues for the authors which should be considered or/and improved:

1.       I have an editorial note here: please ask the authors to carefully read the entire article once again and address the following issues: italicize proper names of species when mentioned, and also please fill in/write complete sentences that end with a period in the middle of a sentence, e.g line 319-325

2.       In my opinion, the research objective (aim of the study) should be formulated more precisely in the introduction, in a way that is understandable even for general readers.

3.       English language editing is necessary before publishing the article.

4.       I have a question for the authors: Why was euthanasia of animals conducted specifically on the fifth day? Line 327

English language editing is necessary before publishing the article.

Reviewer 4 Report

The manuscript by Murshed et al. is well written and conducte. However, before it further processing, please clarify the real contribution of the obtained results to the development of veterinary sciences, and especially the prophylaxis/treatment of coccidiosis in any food producing animal. In addition, the authors must scientifically document the pathogenecity of Eimeria papillate for any food producing animals. For me, this is the biggest concern of the manuscript! These aspects must be clealy mentioned within the first part of the introduction section, and largely discussed, even under the form of study limitation, within the Discussion heading.

Minor issues

L15: “was the control” instead of “is the control

L16: “E. papillate” (e.g. L35 Eimeria, reference list, etc.)– please ensure that the scientific name of the species is written in italics throughout the manuscript

L19: „were sacrificed” – inappropriate formulation, replace it

L28: „our findings” – please avoid the using of personal mode formulations throughout the entire manuscript, it is not so characteristic for the scientific style

Author Response

Review Report 3

Comments and Suggestions for Authors

The manuscript by Murshed et al. is well written and conducte. However, before it further processing, please clarify the real contribution of the obtained results to the development of veterinary sciences, and especially the prophylaxis/treatment of coccidiosis in any food producing animal. In addition, the authors must scientifically document the pathogenecity of Eimeria papillate for any food producing animals. For me, this is the biggest concern of the manuscript! These aspects must be clealy mentioned within the first part of the introduction section, and largely discussed, even under the form of study limitation, within the Discussion heading.

Parasite E. papillate is studied from two aspects

First, because it is a parasite that spreads in mice used in research experiments in the laboratory that are mainly relied upon globally, and therefore it must be treated before using it in research experiments in order to obtain the correct results.

secondly, Mice are the mammalian model that is utilized most frequently in fields of biomedical research all over the world. Consequently, E. papillate is an excellent model for the investigation of poultry coccidiosis, because poultry plays an important role in covering human needs for protein-rich white meat.

Minor issues

L15: “was the control” instead of “is the control”

Done

L16: “E. papillate” (e.g. L35 Eimeria, reference list, etc.)– please ensure that the scientific name of the species is written in italics throughout the manuscript

Done

L19: „were sacrificed” – inappropriate formulation, replace it

Done

L28: „our findings” – please avoid the using of personal mode formulations throughout the entire manuscript, it is not so characteristic for the scientific style

Done
